

# A New Working Fluid for Condensation Particle Counters for Use in Sensitive Working Environments

*Patrick Weber[1,4], Oliver F. Bischof[1, 2], Benedikt Fischer[1], Marcel Berg[1], Jannik Schmitt[1], Gerhard Steiner[3], Lothar Keck[3], Andreas Petzold[1,4] and Ulrich Bundke[1]*

[1]Forschungszentrum Jülich GmbH, Institute of Energy and Climate Research 8 – Troposphere (IEK-8), Jülich, Germany

[2]TSI GmbH, Particle Instruments, Aachen, Germany

[3]GRIMM Aerosol Technik Ainring GmbH & Co. KG, Dorfstraße 9, 83404 Ainring, Germany

[4]Institute of Atmospheric and Environmental Research, University of Wuppertal, 42119 Wuppertal,
Germany

*Correspondence to*: Patrick Weber (p.weber@fz-juelich.de) and Ulrich Bundke (u.bundke@fz-juelich.de)

**KEYWORDS** Condensation Particle Counter, Working Fluid, Aerosol Measurement



**ABSTRACT.** The working fluid of a condensation particle counter (CPC) is one of its essential characteristics. Butanol is commonly used as the working fluid in alcohol-based CPCs due to its proven performance in various applications and a wide range of working conditions over the past almost five decades. At the same time, butanol has several disadvantages including its strong, unpleasant odour, negative effects when inhaled over prolonged periods, and flammability, making it troublesome to use

in all places with strict safety regulations. We are proposing to address these negative issues by replacing it with dimethyl-sulfoxide (DMSO), a substance that has not been used as the working fluid in a CPC up to now. DMSO is an odourless, non-flammable, non-toxic substance that is easily accessible and inexpensive. During thorough experiments, this new substitute working fluid has proven to be stable in its performance in CPC and storage. We could show that DMSO behaves equivalent to butanol when

used to operate a CPC in terms of the instrument`s counting efficiency, D50 cut-off diameter and concentration linearity. In addition, we have demonstrated this for operating pressures ranging from ambient down to 200 hPa without the need for any safety precautions. Our new working fluid was tested on three CPC units representing two different models, almost always in side-by-side measurements. So far, we have achieved the best results with operating temperatures of 40 °C for the CPC's saturator and

5 °C for its condenser. To address a less desirable property of DMSO, it could be mixed with a slight amount of water to decrease its freezing point significantly. When mixed accordingly, the substance stays liquid even in work environments with temperatures ≤ -10 °C and without any impact on the CPC`s counting efficiency performance.

## 1 Introduction

Aerosol science came much more into the focus in recent years. This was driven by the ongoing pandemic (Somsen et al., 2020) and rising awareness of the adverse effects of aerosol particles on



aspects such as climate change, air quality, and public health, and all their interrelations (Von Schneidemesser et al., 2015). Specifically, the monitoring of atmospheric aerosol (Mcmurry, 2000), including extreme environmental conditions like measurements on airborne platforms(Petzold et al.,

2013), exhaust aerosol from various sources (Giechaskiel et al., 2009; Petzold et al., 2011; Bischof et al., 2019) indoor aerosol (Salimifard et al., 2020), and airborne viruses are current key applications of condensation particle counters.

A condensation particle counter (CPC) can measure the aerosol particle number concentration by increasing the size of nanometre-sized particles in a supersaturated environment to optically detectable

droplets. In short, the measurement principle of a common CPC can be expressed in three process steps:

1) saturation, by which vapour is formed from a working fluid inside the flow path of the aerosol sample;

2) supersaturation of the vapour in presence of the sample flow, so that the aerosol particles are subsequently acting as condensation nuclei, get activated (non-equilibrium growth) and are

growing to a detectable size by a factor of several orders of magnitude; and

3) the detection of individual particles through scattered light that occurs when they are passing through an optically focused, incident radiation source. In order to collect the scattered light, a photosensitive element e.g. a photomultiplier or a photo diode (Bischof, 2022) is typically used.

This basic principle dates back to John Aitken, known for his early experiments in which he manually counted particles that had grown due to supersaturation of water vapour (Aitken, 1888). Today, three working fluids are commonly used, namely n-butyl alcohol (or n-butanol), isopropyl alcohol (2-propanol or isopropanol;) and water. For all working fluids, the detection efficiency has been characterised over a specific range of operating pressures (e.g.,(Brock et al., 2000; Bundke et al., 2015; Gallar et al., 2006;

Hermann et al., 2007) which demonstrated the applicability of each working fluid for low-pressure operation CPCs. It should be noted that the use of both butanol and isopropanol is limited by the fact that both are highly flammable liquids. In contrast, water has the advantage to avoid the health and



safety concerns of alcohol. Disadvantageously, water has a three times higher mass diffusion coefficient (Hering et al., 2005; Mei et al., 2021) which increases the consumption of the working fluid during operation and its likelihood of algae growth during longer times of inactivism. Activation of particles is hindered, e.g. by aerosol properties like water solubility and lipophilicity.

Our search for a new, non-toxic, non-flammable, odourless working fluid was motivated by the requirements of aviation authorities that severely restrict the use of instrumentation on passenger aircraft. The European research infrastructure "In-Service Aircraft for a Global Observing System" (IAGOS) (Petzold et al., 2015; www.iagos.org) has been created to monitor all essential climate variables of the atmosphere, including aerosol particles (Bojinski et al., 2014). The monitoring takes place through regular and global-scale measurements conducted on board a fleet of commercial passenger aircraft equipped with automated scientific instrumentation. The IAGOS aerosol package uses a butanol-based CPC as described in detail by Bundke et al. (2015) and provided two years of measurement onboard the IAGOS CARIBIC flying laboratory. It obtained special permission of German airline Lufthansa and its use included the observation of the Raikoke volcanic ash plume. However, the fact that butanol is a flammable liquid strongly hinders the international certification of this instrument for use aboard passenger aircraft.

This study is part of the ongoing development of the IAGOS aerosol instrument, in response to these flight safety aspects. For the next generation of the IAGOS aerosol package, we will operate two butanol condensation particle counters (model 5411 Sky-CPC; Grimm Aerosol Technik, Ainring, Germany, a variant of Grimm model 5410 CPC using aviation-grade components and materials) alongside an optical particle counter (OPC, model 1.129, Grimm). One CPC is dedicated to measure the total aerosol concentration and the other is equipped with a thermodenuder operated at 250 °C upstream to measure only the non-volatile fraction of aerosol particles.

The new working fluid dimethyl-sulfoxide (DMSO; $C_2H_6OS$; CAS-Nr. 67-68-5; 99,9%) that we introduce in this study is hitherto unknown for use in a CPC, but well known within the aerosol community, since it is an intermediate step of the well-known chemical reaction beginning with DMS (dimethyl sulphide).



Its final product is sulphate, which serves as cloud condensation nuclei (Fung et al., 2022). The principal benefit of DMSO for use on passenger aircraft is that it does not require any safety regulations according to the Globally Harmonized System of Classification and Labelling of Chemicals. According to this internationally agreed-upon standard, it does not have any physical or health hazard characteristics. Therefore, this working fluid does not present an obstacle during the certification process for aviation operation. An effect on measuring stations, that observe DMSO and DMSO products, is considered as minimal as the amount of DMSO released in the environment is far less, than the natural production.

## 1.1 Working Fluid characteristics

A working fluid's physical characteristics are important for condensation particle counters' performance. One governing parameter is the maximum saturation ratio, which can be achieved in a specific CPC design. Our first objective was to find a substance with similar physical-chemical properties like butanol or water, which are widely used as working fluids in condensation particle counters. The second motivation was to address any safety restrictions, that make the working fluid suitable for sensitive working environments such as hospitals, schools, public spaces, industrial manufacturing, but also commercial passenger aircraft. DMSO was eventually identified as a candidate substance based on the authors' experience with organic solvents. Besides the advantages of not having any physical or health hazard characteristics, DSMO is also soluble in water. Therefore, the relatively high freezing point of DMSO of 18°C can be adjusted to even -100°C or even below simply by adding defined amounts of water (Havemeyer, 1966).

The supersaturation of the working fluid vapour determines the smallest possible particle size which will be activated to grow, and which can still be detected. For pure liquids this ratio can be expressed through the Kelvin equation (Hinds, 1999; Thomson, 1871):

$$\frac{p}{p_s} = \exp\left(\frac{4\sigma M}{\delta R T d}\right)$$



120   Where the following parameters are used: The surface tension of the liquid σ, the molecular weight

M, the density δ, the general gas constant R, the absolute temperature T and the Kelvin diameter d. For

DMSO, a surface tension of 0.043 N/m, a molecular weight of 0.078 kg/mole and the liquid density of

1100 kg/m$^3$ is used (CRC Handbook, Internet Version 2022).

   The Kelvin diameter shows from which saturation rate $p/p_s$ is in equilibrium with his surroundings and

the droplet neither starts to grow nor shrink to due condensations processes or evaporating.

   When a soluble nucleus is considered, the particle growth can be initiated by a mere percentage

needed for a pure liquid. A given mass of a salt reduces the given vapor pressure at the droplet surface.

The relationship between the kelvin ratio and the particle size with dissolved matter is given by (Hinds,

1999):

$$\frac{p}{p_s} = (1 + \frac{6imM}{M_s \delta \pi d^3})^{-1} \exp\left(\frac{4\sigma M}{\delta RTd}\right)$$

   Where m is the mass of the dissolved material and $M_s$ the molecular weight. The constant i is the

number of molecules the salt will dissolved into. This ratio can then be solved for a particle size when

the supersaturation ratio is known. To calculate the saturation vapour pressure, the empirical Antoine

equations can be used (Antoine, 1888):

$$log_{10}p = a - \frac{b}{c + T}$$

For which a, b and c are component-specific constants, p is the vapour pressure and T the

Temperature. For this is to be solved for two different temperatures, the ratio of it will be the

supersaturation for this calculated temperature difference. For DMSO, the component-specific

constants are a= 5.23; b= 2239.2; and c= -29.2 (Domalski and Hearing, 1996).




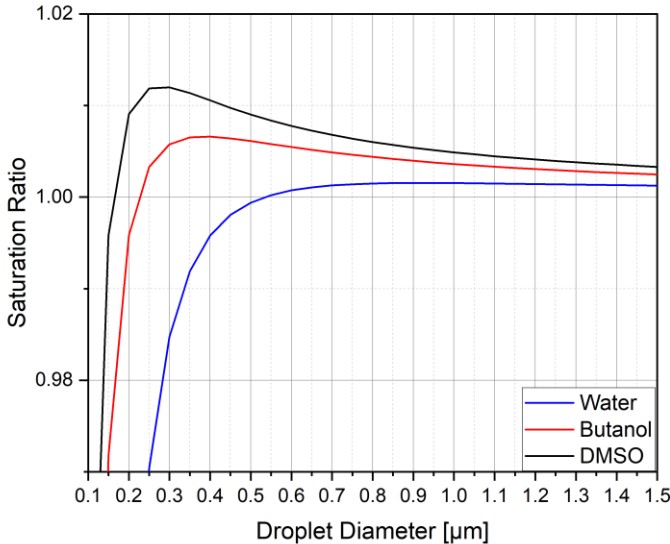

**Figure 1. Saturation ratio of different working fluids in dependence of the droplet diameter with a starting diameter of sodium chloride of 13 nm (Hinds, 1999).**

To understand the influence of the working fluid on the activation process inside a CPC, the process can be summarised as follows. The vapour pressure above the curved particle surface is described by the Kelvin Köhler equation (Thomson, 1871) as visualized in Figure 1. Here the saturation ratio is defined by the ratio of the actual vapour pressure to the vapour pressure above a flat surface at a given temperature. Particles smaller than the critical diameter grow or shrink in equilibrium with the vapour

in their vicinity. As the particles pass the critical diameter as the critical saturation ratio is exceeded in their vicinity, they will grow in non-equilibrium if they face saturation ratios above 1 condition, which is defined as supersaturation. This process is known as particle activation. The critical supersaturation depends on the initial size of the aerosol particles and will be higher for smaller particles (Hinds, 1999). As shown in Figure 1, the saturation ratio necessary to activate a sodium chloride particle of an initial

size of 13 nm is at 1.012 for DMSO, 1.007 for Butanol and 1.002 for Water. The necessary supersaturation for activation increases with decreasing particle size. Thus e.g. for particles size of 6 nm a saturation ratio of 1.037, and for 3 nm a saturation ratio of 1.078 has to be exceeded for DMSO.



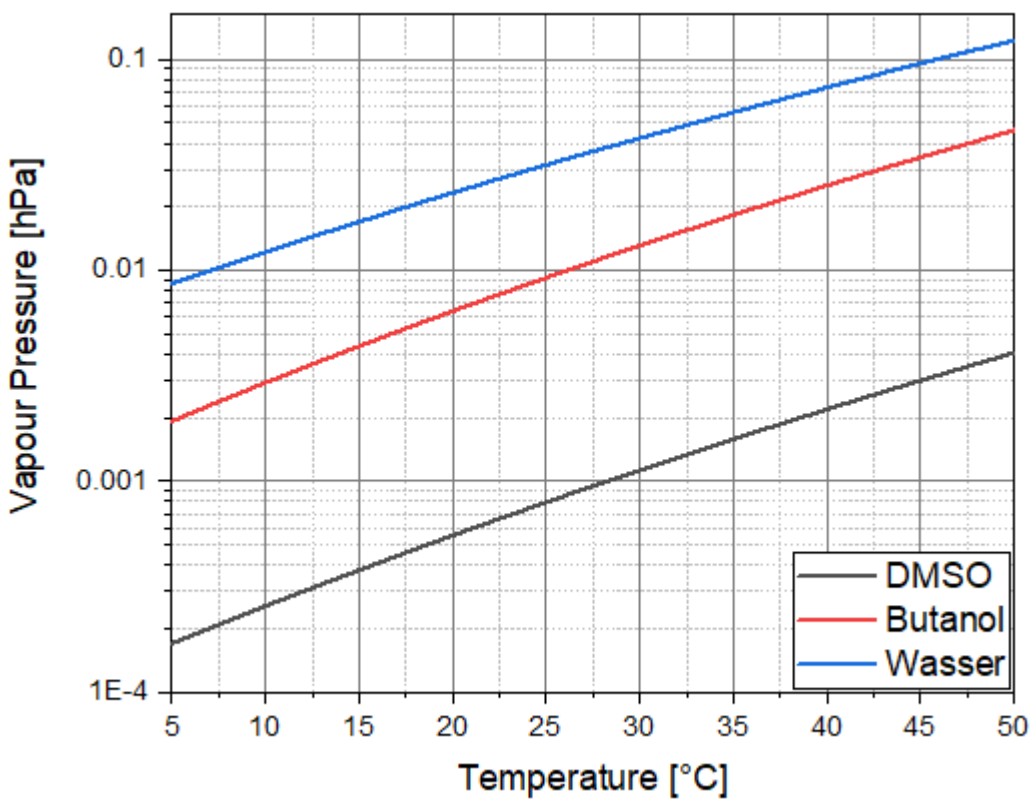

**Figure 2. The theoretical saturation vapour pressure of butanol, water and DMSO as calculated by the Antoine equations(Antoine, 1888; Gutmann and Simmons, 1950).**

As can be seen in Figure 2, the saturation vapour pressure of DMSO is far below that of butanol, but with a similar slope of 0.031, while it is 0.026 for water. This indicates the same supersaturations could be achieved with the same temperature difference between the saturator and condenser of
condensation particle counters, but with a much lower consumption of the working fluid. For DMSO, a supersaturation of about 10% (saturation ratio of 1.1) could be achieved in this work with a saturator temperature of 40 °C and a temperature drop to 5 °C in the condenser part of the CPC unit used.

## 2 Methods

In the experimental set-up shown in Figure 3, the sample-line pressure is regulated using the flow mass balance using mass flow controllers (MFC). The test aerosol is size classified by a differential



mobility analyser (DMA, Model M-DMA 55-U, Grimm) while a faraday cup electrometer (FCE, Model 5.705, Grimm) serves as a reference instrument for the particle concentration. One Sky CPC 5.411, which from now on we designate as B-CPC to clarify that it was operated with butanol as its working fluid, and

a second Sky CPC 5.411, which we will refer to as D-CPC to highlight that it was operated with DMSO. In parallel, a CPC 3772-CEN (TSI Inc., Shoreview, MN, USA) that was also performed with DMSO was investigated as well. A more detailed description of the experimental setup is provided in prior studies (Bundke et al., 2015; Bischof, 2022). To provide a steady and constant particle production from a NaCl-water solution, a continuous output atomiser (COA, model 3076, TSI Inc.) was used (Liu and Pui, 1975).

After the nebulised aerosol flow passes through a diffusion dryer tube, it moves through an aerosol neutraliser equipped with a radioactive Am-241 source. Subsequently, a monodisperse aerosol flow is generated utilising a Vienna-type Differential Mobility Analyzer (DMA, model M-DMA 55-U, Grimm). This monodisperse aerosol is guided to the low-pressure zone by passing through a critical orifice. The aerosol flow is then diluted in a controlled manner within the mixing chamber. The mixing chamber is

also used as a buffer volume to smoothen the pressure regulation controlled by the mass balance. The complete test setup is operated by a LabVIEW (National Instruments Corp., Austin, TX, USA) program that was custom-made in our laboratory. Here several mass flow controller function as controlling elements with a PID approach. Additionally, the relative humidity can be actively controlled by adding a stable humidified air flow into the mixing chamber, which is limited to approximately 30% relative

humidity. After passing the mixing chamber, the aerosol flow is provided to the measuring instruments along a common sampling line. An individual isokinetic, iso-axial sample inlet in the centre of the sample line guides the aerosol flow to each device.

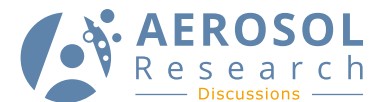

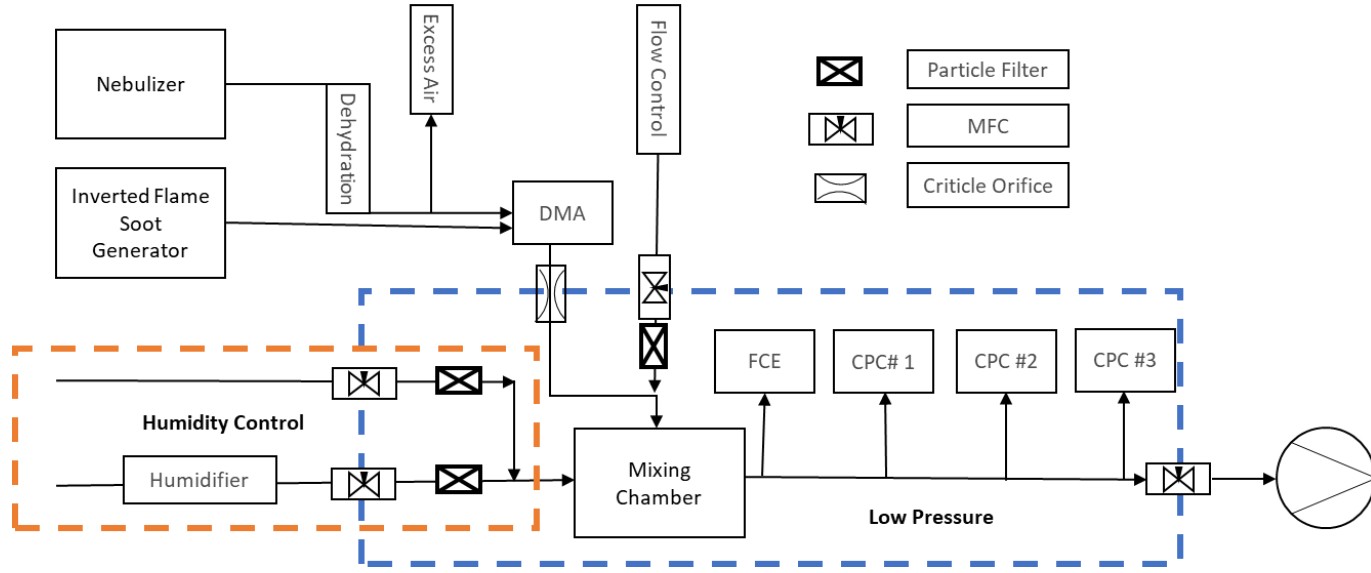

**Figure 3. Flow schematic of the laboratory set up for the low-pressure characterisation with two aerosol sources.**


During our experiments, the DMA was operated stepwise for 30-second periods to avoid time delays and undefined transition states of aerosol characteristics in the different instruments. Each voltage level corresponds to different particle sizes starting at an upper limit of 140 nm and going down to 2.5 nm in

diameter. To avoid transition effects and to achieve an equally distributed aerosol inside all measuring instruments, the first 15 seconds for each particle size setting of the DMA were excluded from the dataset used for the final analysis.



An inverted flame soot generator (Argonaut Scientific Corp., Edmonton, AB, Canada) was used as a second aerosol source. For best performance, the soot generator was operated with an oxidation-air-to-propane ratio of 7.5 L/min air to 0.0625 L/min propane. Previous studies have determined that this flow ratio ensures stable aerosol production with low organic carbon soot (Bischof et al., 2019; Kazemimanesh et al., 2018).

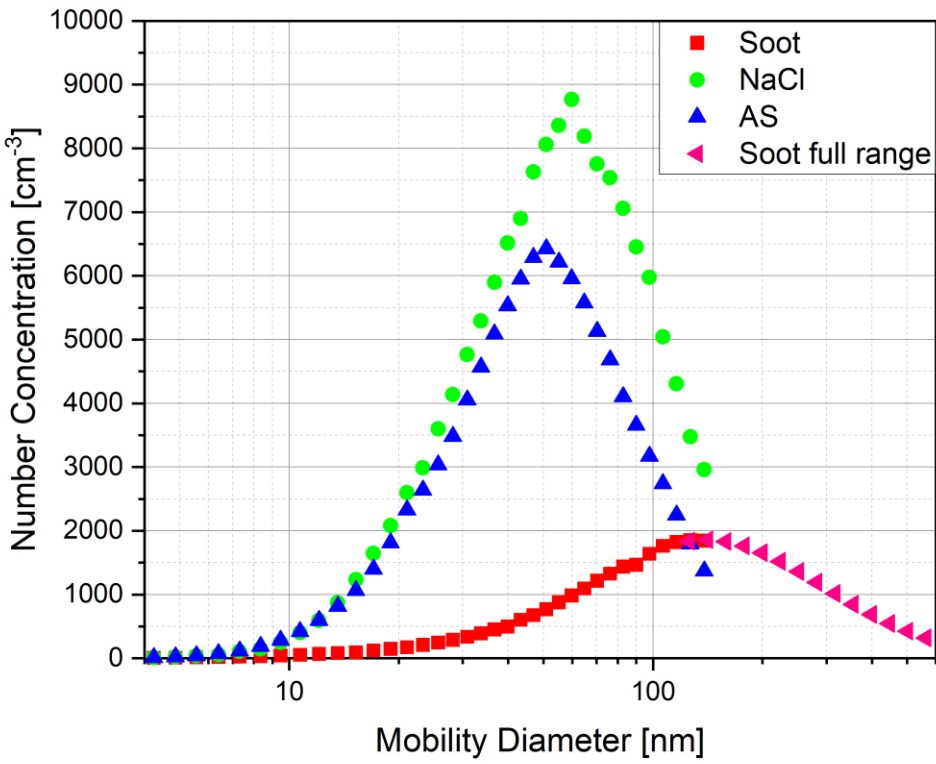


**Figure 4.** Particle size distributions of ammonium sulphate (AS), sodium chloride (NaCl) and fresh combustion soot as measured by the combination of DMA and electrometer.

     As mentioned earlier, the atomiser was used to nebulise salt solutions. Sodium chloride (NaCl) and ammonium sulphate were both used for this purpose. Additionally, fresh combustion soot was

measured. In Figure 4, the corresponding size distributions for these aerosol types are shown. The DMA settings were chosen, to provide the greatest possible size resolution at the smallest particle sizes near




the range of the expected cut-off diameters. This resulted in a maximum particle mobility size of 138 nm on the upper end. Complete particle size distributions were taken from (Weber et al., 2022).

**2.3 Data analysis procedure**

An important aspect of measuring nanometre-sized particles by the DMA technique is the misclassification of larger multiply charged particles leaving the DMA with the same mobility as the desired singly charged ones. This effect leads to a notable difference in the counting rate between a condensation particle counter and an aerosol electrometer. To address this artefact, the correction

procedure introduced by Bundke et al. and later on adapted by Bischof (Bundke et al., 2015; Bischof, 2022) was used.

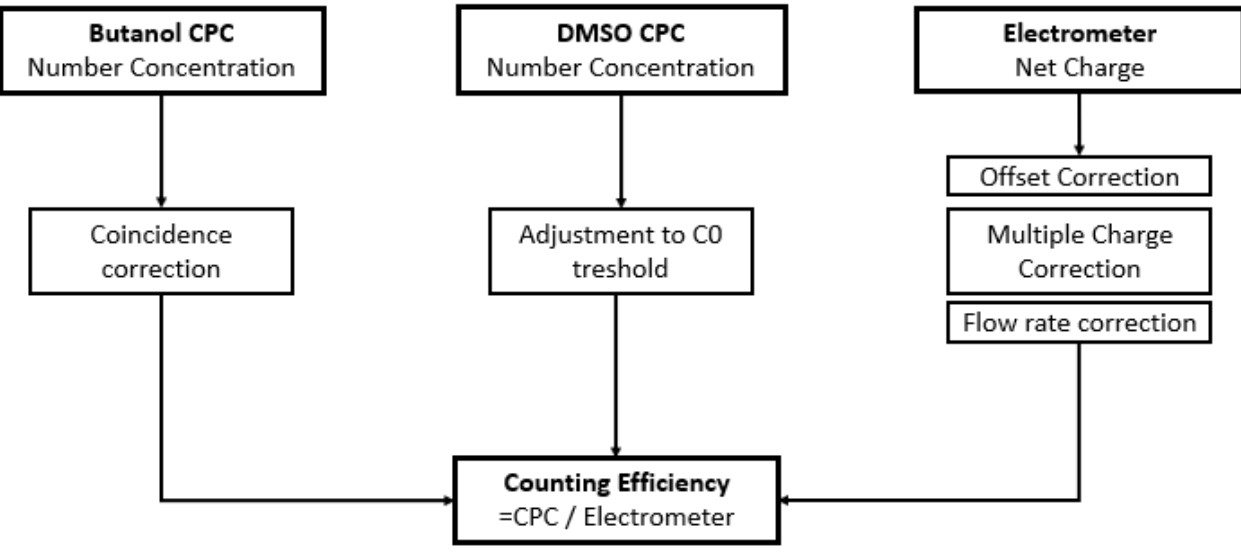

**Figure 5. Flowchart of the data inversion procedure for particle concentration determination.**




The Multiple charge correction is calculated with a correction scheme, that is based on measured size distribution from the condensation particle counter and the electrometer in accordance to DMA theory by Bundke et al. (2015)

The SKY CPC 5.411 reports an internal diagnosis of the particle growth which is displayed as the C1/C0 value. The C1/C0 correction is implemented by simply dividing through the C1/C0 value. This value was reported by the Sky CPC 5.411 as an internal quality parameter. In Figure 6, signals of different droplet sizes are illustrated, that can surpass certain detector thresholds. By dividing by C1/C0, the CPC counts all particles that reach the first detector threshold. DMSO droplets smaller than 2.5 µm will therefore not be counted.

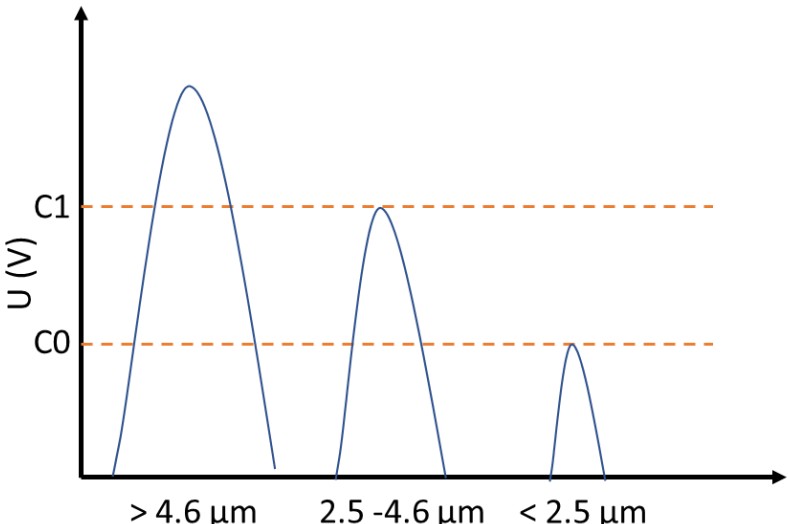


**Figure 6. Illustration of the pulse height of DMSO-Droplets of different sizes with corresponding detector threshold levels.**

The very need for this correction is due to the low vapour pressure of DMSO. If less material is available for condensation growth, the droplets cannot grow. The diameters of the DMSO droplets, which correspond to the signal heights of C0 and C1, are 2.5 − 2.9 um and 4.0 − 4.6 um. The values were estimated from the signal heights of latex test particles, a model for scattered light intensities as a function of particle diameter for the GRIMM measuring cell, assuming spherical homogeneous particles, a wavelength of 660 nm, linear polarization, and refractive indexes of 1.59 + 0.0i for latex and 1.47 + 0.0i for DMSO.



To parameterise the efficiency curves, an exponential fit function introduced by Wiedensohler et al.
(2018) was applied

$$\eta = A\left(1 - \exp\frac{(B - Dp)}{(C - B)} \ln 2\right) \ .$$

Here, $\eta$ is the counting efficiency, $D_p$ is the particle size, and A, B, C are fitting parameters calculated using a multi-parameter fit (Banse et al., 2001; Wiedensohlet et al., 1997) where C corresponds to the $D_{50}$ cut-off diameter.

The fitted function was then used to robustly calculate the efficiency for the cut-off of $D_{90}$.

## 3 Results

During initial experiments we used spare DMSO and applied the substance to a completely dried Sky CPC 5411. To test this, the dried-up CPC performed a test run in the measuring set-up and reported zero counts. The new working fluid was applied and worked as intended and all CPC internal fluid controls were operational like with butanol. We operated this substance with the same parameters as the structurally identical butanol Sky-CPC, which was already in the set-up. For the first run we applied a particle filter to the inlet to check for homogeneous nucleation. No particle counts were observed during zero particle air measurements, independent of instrument temperatures, humidity, or pressure levels. As we started aerosol measurements with pressure levels below 700 hPa, the C1/C0 decreased to 0.9 and dropped further down, as soon the pressure decreased. The C1/C0 reports the ratio of counts that have a higher and a lower detection threshold, used as internal CPC check for sufficient particle growth (described in Fig 6.). A value below 1 indicates, that the particles give a lower scattering signal due to a lower particle size or overall light scattering behaviour. Only particles reaching both signal threshold levels are counted. As a consequence, we increased the saturator temperature from 36°C to 40°C and decreased the condenser temperature from 10°C to 5°C. With this procedure, the C1/C0 value stayed at 1 down to a pressure level of 250 hPa and only decreased slightly to 0.95 at 200 hPa. All values given as number concentration from the Sky-CPC for DMSO values are corrected by the division with the C1/C0





value to report all particles at the lower counting threshold. The very first results were taken with

ammonium sulphate particles. The $D_{50}$ cut-off diameter is used as a primary criterion, as it describes the

particle size, where only 50% of the particles are detected using a reference method. In Figure 7 it is

visible, that the DMSO shows identical $D_{50}$ cut-off diameter as the equivalent instrument using Butanol.

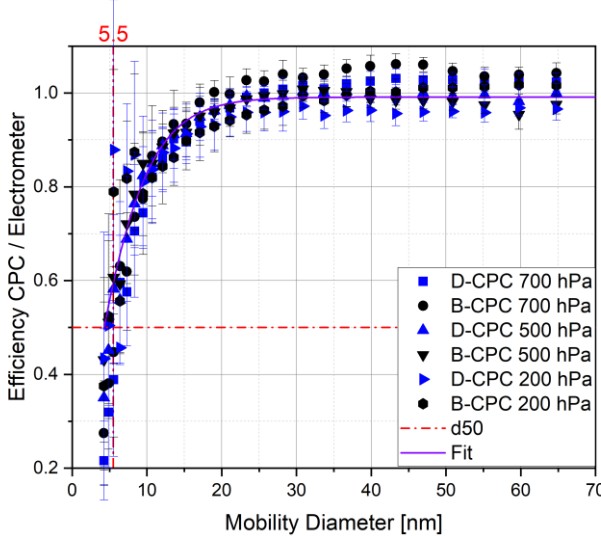

**Figure 7. Counting efficiency curves of the Sky CPC 5.411 operated with butanol (B-CPC) and DMSO (D-CPC) at different operation pressures using ammonium sulphate particles.**

As second step, we tested the new working fluid in a CPC of a different kind. For the CPC 3772-CEN the

main operating principle is the same, but there are slight differences to the Sky-CPC. The Sky-CPC has a

horizontal wick and a continuous draining of condensed water/butanol mixture located downstream of

the saturator working fluid reservoir. The 3772-CEN has a vertical wick, where condensed water may

flow through before reaching the fluid reservoir. This different design will explain some different

behavior observed during long-term experiments described later in the text. Nevertheless, we dried the

3772-CEN for over two days till no particle count were reported and applied the new working-fluid as

well. We changed the temperature parameters equivalent to the DMSO Sky-CPC with 40°C at the

saturator and 5°C at the condenser. Therefore, the 3772-CEN was not operated as intended by the



European Committee for Standardization (CEN). As visible in Figure 8 counting efficiencies for all CPC do not significantly differ from each other, regardless of the instrument and applied pressure.

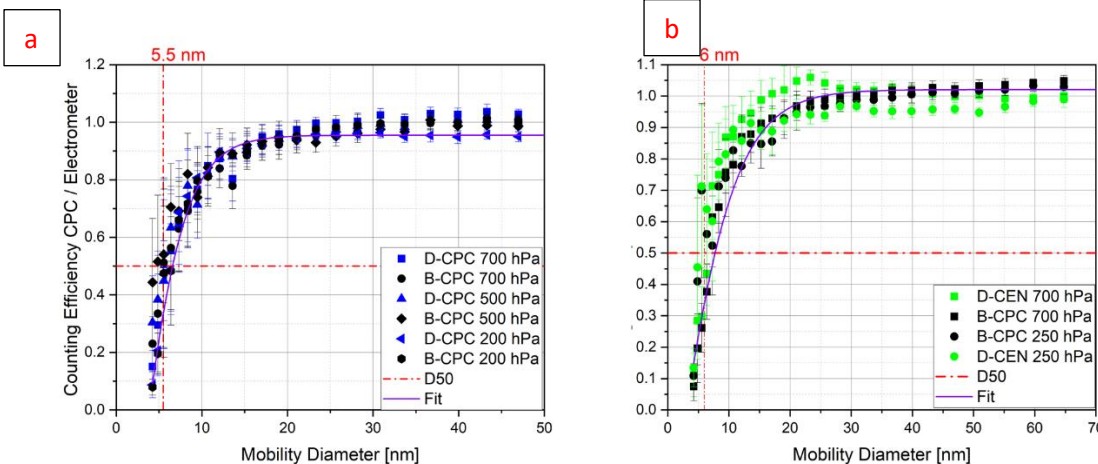

**Figure 8. Counting efficiency curves with respect to the electrometer reference instrument and corrected for multiple charge of the Sky CPC 5.411 (a) as well as the CPC 3772-CEN (b), both operated with butanol (B-CPC) and DMSO (D-CPC; D-CEN) at different operating pressures using sodium chloride particles.**

To test another particle type, we chose fresh combustion soot aerosol, as it is a non-soluble substance and is not hydrophilic. The counting efficiencies do not differ as is visible in Figure 9 like in the previous experiments.

We repeated all experiments using a mixture of 10% volumetric water to 90% DMSO (DW-CPC) does not influence the measurements in any way see Figure 9. We observed, that the C1/C0 value decreased to below 0.2 for about a minute as soon we reached pressure levels below 250 hPa, indicating, that water is evaporated and influences the particle growth. After a span of one minute, the C1/C0 value rapidly



increased to over 0.98. Using this mixture, the environmental operational conditions range is extended

to -10°C. The pure DMSO substance has a melting point of 18°C.

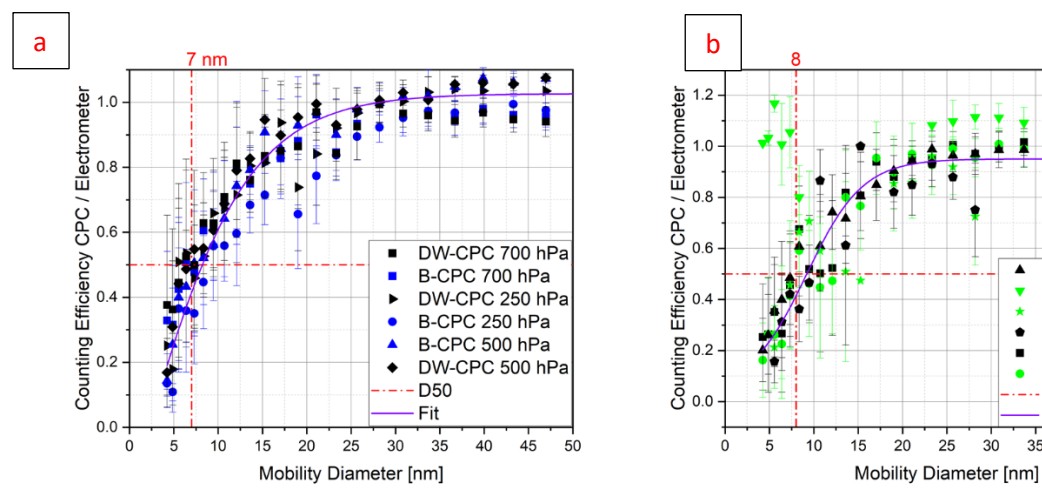

**Figure 9. Counting efficiency curves with respect to the electrometer reference instrument and corrected for multiple charges of the**

**Sky CPC 5.411 (a) as well as the CPC 3772-CEN (b), both operated with butanol (B-CPC) and DMSO (D-CPC; D-CEN) at different**

**operating pressures using fresh combustion soot particles.**

The counting efficiency for soot particles around 5 nm shown in Figure 9 for 700 hPa D-CEN

measurement experience a sudden increase. This need further investigation and could not been

explained.

We tested the substance in our measurement set up at various conditions, as such we increased the

inline humidity to 30%, as we normally measure below 5% Even then, the counting efficiencies as well

as the overall behaviour did not change. Regarding the cut-off diameter for $D_{50}$ and $D_{90}$ the Instruments

do not differ significantly. In Table 1 the fitting parameters and the derived values for the cut-off

diameter are reported for the lowest pressure level at 200 hPa and 250 hPa.



**Table 1. Coefficients of the Exponential Fit of the Counting Efficiency Curves and the D50, D90 cut- off diameter for 200 and 250 hPa for different particle types for the Sky CPC 5.411 operated with Butanol (B-CPC) and DMSO (D-CPC) and CPC 3772-CEN operated with DMSO (D-CEN) are reported.**

| Particle Type | Instrument | A | B | C [nm] | $D_{90}$ [nm] |
|---|---|---|---|---|---|
| Sodium Chloride | B-CPC | 0.99 | 3.9 | 6.7 | 13.6 |
| Sodium Chloride | D-CPC | 0.96 | 3.9 | 6.3 | 13.5 |
| Sodium Chloride | D-CEN | 1.02 | 1 | 6.7 | 18.6 |
| Ammonium Sulphate | B-CPC | 0.98 | 2.1 | 4.8 | 11.9 |
| Ammonium Sulphate | D-CPC | 1.02 | 2.8 | 6.5 | 14.2 |
| Fresh Combustion Soot | B-CPC | 0.95 | 3.4 | 6.8 | 17.8 |
| Fresh Combustion Soot | D-CPC | 1.02 | 2.7 | 5.6 | 11.7 |
| Fresh Combustion Soot | D-CEN | 0.95 | 3.7 | 7.5 | 19.8 |

After the characterization in the lab we conducted several measurements of outside air at the campus of the Forschungszentrum Jülich. The longest measurement lasted 5 days in a row, with the Sky-CPC measuring the same number of particles. We observed that the 3772-CEN had issues during the measurements caused by DMSO. Here the valves for draining and priming got blockey by an swollen gasget. First of all, we needed to extract excess fluid from the device manually. Since the overall

humidity was ranging between 60 and 80%, it is certain that water condensed at the condenser and dropped into the wick of the 3772-CEN, weakening the efficiency. Figure 10 shows at the right panel that starting at minute 60 till minute 80, manual draining was performed. After that, the 3772-CEN



reaches the same concentration levels as the other two CPC. The observation was made with recorded

working fluid consumption: the Sky-CPC was filled with 5 ml of DMSO, whereas the second Sky-CPC was

filled with 20 ml butanol. We run those measurements overnight and as the Butanol filled CPC run dry

in the late morning (see green curve in Fig. 10b), the DMSO filled CPC was still running and counting. At

a first ambient measurement approach, the C1/C0 value of the DMSO filled Sky-CPC stayed at 1 and

decreased slowly until it circulated around 0.8. At a second approach we added a Water/DMSO mixture

and the C1/C0 value decreased towards 0.6 slowly going up again. Anyhow the D-CPC number

concentration was corrected with housekeeping data of the C1/C0 value, which was ranging around 0.8

in Figure 10.

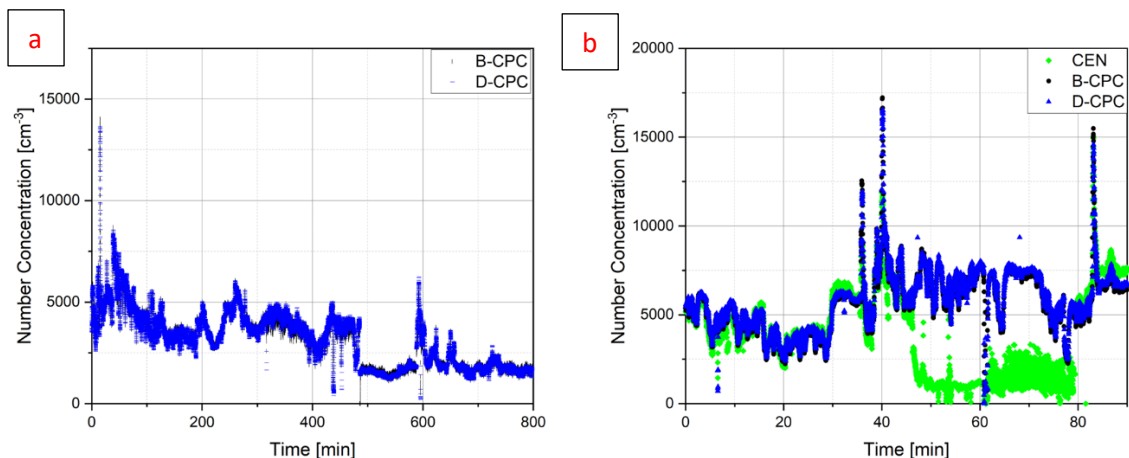

**Figure 10. Ambient air as measured by the Sky CPC 5.411 operated with (a) Butanol (B-CPC) and DMSO (D-CPC) as well for the (b) CPC 3772-CEN operated with DMSO (D-CEN), that needed manual extraction of condensed water, which caused the number concentration**
**to drop.**

As is visible in Figure 11 measurements using DMSO working fluid is not distinguishable from the ones

performed with Butanol. The overall linearity for the new working fluid is shown in Figure 11. The slope

of the linear correlation is 1.01 ±0.02 (r²>0.99) for laboratory measurements as well as ambient air

measurements. For atmospheric measurements we used the B-CPC as well-known reference.




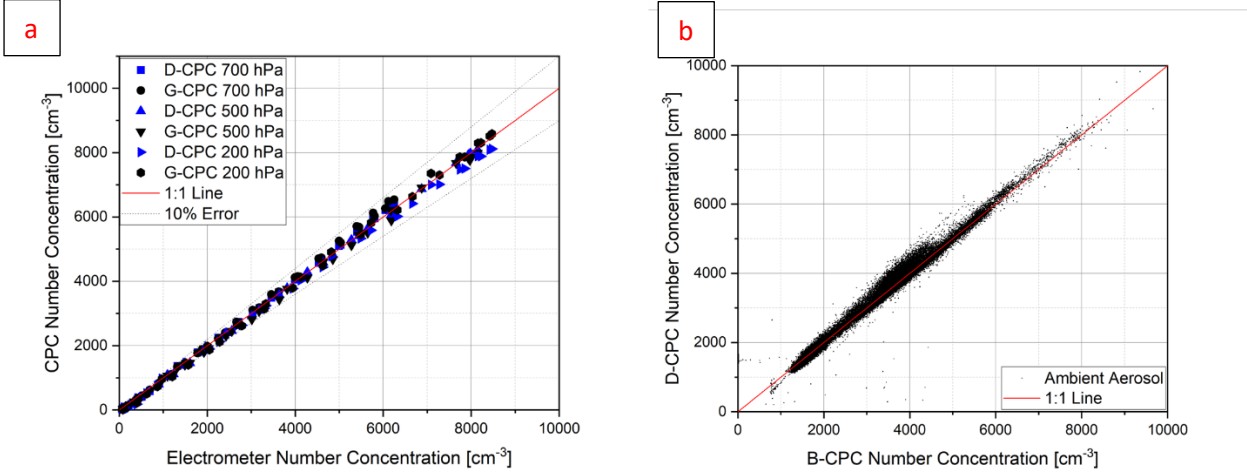

**Figure 11. Comparison of the concentration linearity of two units Sky CPC 5.411 to the electrometer reference at different pressure levels for ammonium sulphate (a) and ambient air (b).**

## 4 Precautions

As already mentioned, the CPC 3772-CEN needed manual draining and the Sky CPC 5.411 needed manual priming. This is because rubber parts soak themselves with DMSO and swell, so they increase in size. With this behaviour, the O-rings expanded inside the valves for refilling and could no longer perform as intended, although it took several days to weeks before the valves stopped working. In order to address this issue, we exchanged those parts with silicon O-rings. We operated the instruments for at least three months with the new working fluid and apart from the valves, no essential parts were affected since the device kept working. We repeated the process with a third Sky CPC 5.411 and it performed the same test as in the first experiments, including the soaked O-ring. As can be in Figure 12, the O-ring, as well as the stamp of the valve, increased by about 1 mm in size.

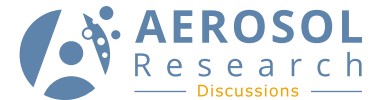

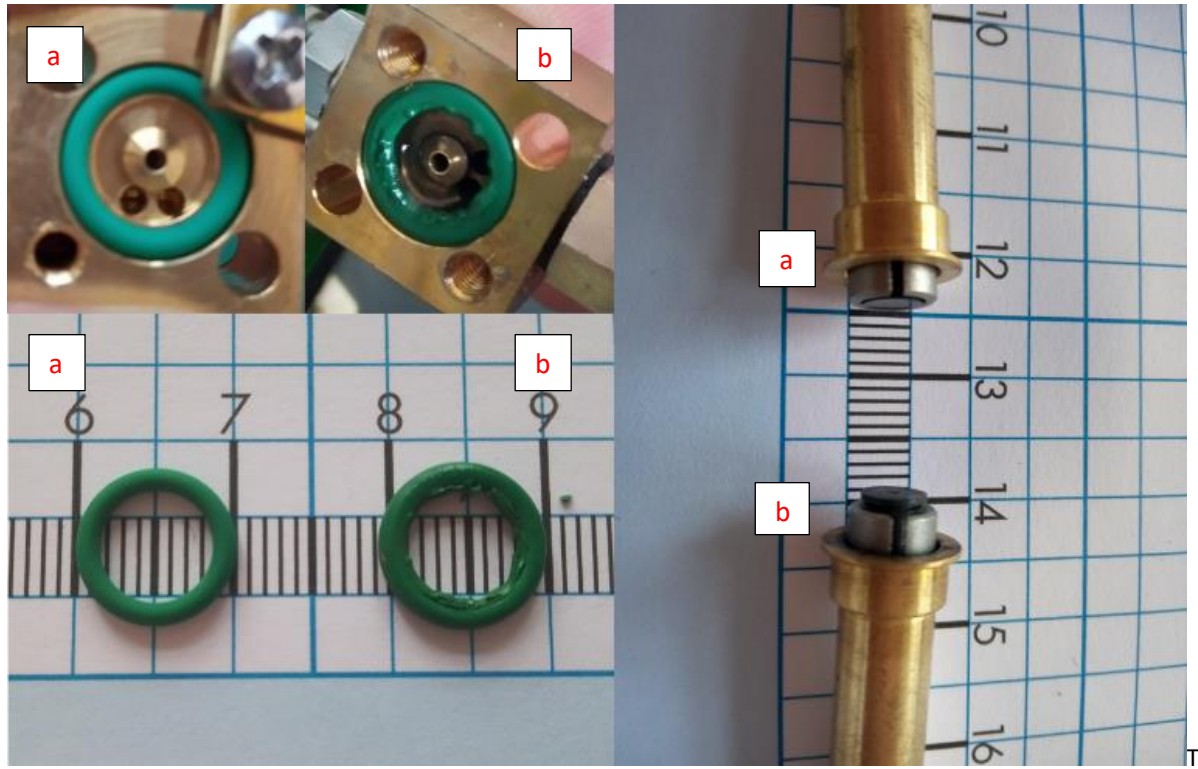

**Figure 12. Picture of the gasket of a butanol CPC and a DMSO CPC (left), used in butanol (a) and DMSO (b) as well as of the stem of the liquid supply valve (right).**

With this increase in size, the sealing ring blocked the stamp and prevented the further flow of liquid to the wick. The continuous pump for draining in the Sky CPC 5.411 was still working as intended.

Cross-sensitivities between DMSO and several reactive atmospheric substances were not tested. We tested for several weeks atmospheric aerosol and did not observe any significant shift between the reference butanol CPC. However, higher ozone or nitrogen oxides concentrations could trigger

secondary organic aerosol activation within the instrument.

In the CPC 3772-CEN, one of several critical orifices in the supply line of the working fluid started leaking and had to be replaced. It is not conclusive that this effect should also be attributed to the use of DSMO, as the instrument's maintenance state or operational errors could also have been factors.

DMSO can bring chemicals into the body by transporting dissolved chemicals through the skin. Bacteria

can convert DMSO into dimethyl sulphide. Therefore, it should be avoided to be disposed into the sewer (Tucker, 2014).



Another precaution that should be mentioned here is that DMSO should not be heated to temperatures of over 150°C, since at that threshold it will lose its thermal stability. When exceeding 150 °C, DMSO slowly starts to degrade into its by-products, which may have different characteristics.

We are not certain if the signal is generated by droplets or solid crystals, because the freezing point of DMSO is below the condenser temperature.

From an instrument manufacturer's point of view, DMSO seems to be an interesting alternative working fluid to the commonly used butanol. Furthermore, at first glance, the modifications required for the correct and safe operation of a Grimm 5411 Sky-CPC appear to be relatively straightforward. However,

as demonstrated in the experiments shown here, there is a need to adapt the CPC design to the specific properties of DMSO, specifically material durability. Also, the detailed condensation process so far remains an open question that needs to be addressed in future studies in order to adapt and optimize the particle detection in upcoming CPC models.

Equally important, the benefits of using DMSO as a substitute working fluid by far exceed the effort

needed to prepare a CPC for its use also from an instrument manufacturer's perspective. In TSI's most recent generation of butanol-CPCs, there are only few places where DMSO comes in contact with materials that it could degrade or swell quickly. In practice this means that those O-rings and gaskets that are by default made of polycarbonate (PC), polyvinylidene fluoride (PVDF) like Kynar®, or polyvinyl chloride (PVC), will need to be exchanged by their silicone equivalents. While this is not technically

difficult, the exact replacement should be assessed and approved separately for every CPC generation and model by the manufacturer. Also, all modifications will need to be tested over much longer periods of time than the brief test period in this work as components such as the instrument optics might get impacted only after several weeks or months of exposure to DMSO. Of course, this would change with the introduction of a next-generation of alcohol-based CPCs that are designed for use with DMSO from

the beginning.

**5 Conclusions**

In this work, we have introduced dimethyl sulfoxide (DMSO) as an appropriate new substance to replace butanol as the working fluid in alcohol-based condensation particle counters (CPC), which is an odourless, non-flammable and non-toxic substance. We would like to emphasize that DMSO overcomes the notable health and safety concerns that arise when using butanol in sensitive working environments, and this fact alone makes it a desirable candidate for aircraft operation. Secondary benefits include the lower consumption when operating CPCs and its lower cost.

In our experiments we have demonstrated that the two different CPC models that we tested continued to operate as expected under several operational conditions. When changing the operation pressure from ambient level down to 200 hPa, which corresponds to the cruising level of passenger aircraft used within the IAGOS research infrastructure, the DMSO-operated CPC units displayed the same counting efficiencies as the butanol-operated CPCs. Even in very humid environments with a relative humidity of around 80%, the CPC operated with DMSO performed equivalent to a butanol CPC. It should be noted that at those high levels of humidity, the CPC 3772-CEN had to be drained frequently, while the Sky CPC 5.411 had to be corrected using its internally recorded C1/C0 value. Nevertheless, for all pressure levels and particle types investigated, the DMSO-operated CPC performed essentially identical to the equivalent butanol-operated CPC. Further investigations with different temperatures for saturator and condenser are planned.

The introduction of DMSO as the working fluid requires no significant modifications of a CPC's hardware apart from minor changes to its liquid supply valve and a replacement of any gasket that comes in contact with the working fluid. Our initial results also indicate that the lower detection limit of the CPC might even be smaller with the new working fluid as a higher supersaturation can be achieved. This is due to the fact, that DMSO has a far higher flaming point than butanol. This will be investigated further in cooperation with CPC manufacturers by using a custom-designed saturator block, heating and reservoir.



*Acknowledgements*. Parts of this work were supported by the German Ministry of Research and
Education in the joint research project IAGOS-D (Grant Agreement No. 01LK1301A), and by the HITEC
Graduate School for Energy and Climate at Forschungszentrum Juelich. A special thanks to Dr. Jürgen
Dornseiffer for providing the very first amounts of liquid we needed to perform initial tests.

*Contributions of co-authors.* PW and UB had the idea, PW performed all instrument maintenance,
experiments and data analysis. PW and UB, OB performed the instrumental set-up. UB, PW and BF
designed the LabVIEW environment of the experimental set up. MB and JS helped during instrument
preparations. LK calculated the droplet size for the threshold detection. PW, OB, GS, UB and AP
contributed to the manuscript and the interpretation of the results.

*Conflict of interest.*
OFB is a full-time employee of TSI GmbH, a subsidiary of TSI Inc., which has a potential direct or indirect financial
interest in the subject matter discussed in the manuscript.
GS and LK are full-time employees of Grimm Aerosol Technik Ainring GmbH & Co. KG, which has a potential
direct or indirect financial interest in the subject matter discussed in the manuscript

The other authors declare that they have no conflict of interest.

*Pending Patent*

**Weber, P., Bundke, U., etc. Verfahren und Vorrichtung zur Vergrößerung von Aerosolpartikeln, German
patent PT 0.3349/pa, filed November 23, 2022.**

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
