# Peer review of "A New Working Fluid for Condensation Particle Counters for Use in Sensitive Working Environments"

_Aerosol Research, 2023_

## Author Comment (AC1)

The manuscript by Weber et al. introduces dimethyl sulfoxide as a CPC working fluid. Existing commercial CPCs are adopted to operate with DMSO, the CPCs are calibrated using soot, NaCl and AS at several pressures intended for aircraft measurements, and the CPCs were operated side by side to sample atmospheric aerosol for few days. The idea of using DMSO as a working fluid is certainly interesting for the aerosol community, the experiments appear well made and reliable, and the authors were very generous in also giving out lots of practical details about modifying the existing CPCs for DMSO. I have only very minor suggestions that the authors may consider adding to the manuscript final version.

- Intro, there are few previous attempts to use other liquids than butanol, propanol or water as a CPC working fluid. One paragraph could be added to give the reader an idea about the other options that have worked in the past, as this is not the first try

  **Answer: In L63 we named several common working fluids. Nevertheless, we will add some information in the introduction:**

  **Dibutylsebecat/DOS/DEHS; No GHS- Symbols, but only suitable for high temperature applications (Kupper et al. 2020)**

  **Diethylene Glycol (DEG); Glycin: Great application for sub nm particles, but droplets only grow to very small sizes, so a "Booster" CPC is needed (Iida et al. 2009)**

  **Theoretical approach for working fluids mostly alkene, alcohols organic acids and several aromatic compounds, but does not include sulfuric solvents (Magnusson et al. 2003)**

- L67, alcohols also smell when not vented properly.

  **Answer: We stated in the abstract: "…butanol has several disadvantages including its strong, unpleasant odour, negative effects when inhaled over prolonged periods, and flammability, making it troublesome to use in all places with strict safety regulations."**
  **We will also add (or move?) this information in (to) L67.**

- Fig2, wasser

  **Answer: Iam sorry, my fault. We will change this.**

- L266 forward, here you describe the meaning of the C1/C0, ie. that they are the counts above these thresholds. please add it to the previous section where the correction is introduced.

  **Answer: We will add a phrase at L234. "The particle number concentration reported by the Sky-CPC is divided by the C1/C0 value. The reported particle**

**number concentration is the number of particles that deliver a scattering signal when passing both detection threshold levels C1 and C0. By dividing the reported number concentration by C1/C0, the number concentration is adjusted to the C0 threshold […] The diameters of the DMSO droplets, which correspond to the signal heights of C0 and C1, are 2.5 – 2.9 um and 4.0 – 4.6 um respectively to the different thresholds**"

- Table1, please include all data here, also the curves at normal pressure are interesting

  **Answer: Our measurement set-up is not suitable to report cut-off efficiencies for normal (1000 hPa) pressure as the Aerosol sample flow is limited by a critical orifice. Without a deltaP, the amount of aerosol entering the measurement line is too low. We could add the information for the curves for 700 hPa, but as visible in the figures: the curves are nearly indistinguishable; therefore the data does not change significantly. We will add this information in the text.**

- Section 4, I commend the authors for giving these practical details

  **Answer: Authors: 😊 😊😊😊😊😊😊😊😊**

References:

Lars-Erik Magnusson, John A. Koropchak, Michael P. Anisimov, Valeriy M. Poznjakovskiy, Juan Fernandez de la Mora; Correlations for Vapor Nucleating Critical Embryo Parameters. *Journal of Physical and Chemical Reference Data* 1 December 2003; 32 (4): 1387–1410.

Kenjiro Iida , Mark R. Stolzenburg & Peter H. McMurry (2009)
Effect of Working Fluid on Sub-2 nm Particle Detection with a Laminar Flow Ultrafine Condensation Particle Counter, Aerosol Science and Technology, 43:1, 81-96, DOI: 10.1080/02786820802488194

Martin Kupper, Martin Kraft, Adam Boies & Alexander Bergmann (2020) High-temperature condensation particle counter using a systematically selected dedicated working fluid for automotive applications, Aerosol Science and Technology, 54:4, 381-395, DOI: 10.1080/02786826.2019.1702920

---

## Author Comment (AC2)

Weber et al "A New Working Fluid for Condensation Particle Counters for Use in Sensitive Working Environments"

This paper presents the possibility of using a different operating fluid in existing CPC instruments to replace the more commonly used butanol. This is important because butanol is expensive to purchase, expensive/difficult to ship because of flammability and has known health issues. They show that the proposed substitute (dimethyl-sulfoxide, DMSO) can provide similar counts to those acquired using butanol but may require some minor changes to the instrument (changes in temperature settings and some fitting replacement).

My comments are primarily technical - related to use of English, but there are a couple places where further clarity could be provided. The comments are given in order of appearance. I would recommend that the authors find a native English speaker to review the paper prior to resubmission.

**Answer: Thank you! We appreciate your efforts to improve the manuscript.**

L40 "Aerosol science came much more into the focus in recent years." change to "Aerosol science has come more into focus in recent years."

**Answer: Ok**

L43 - seems like there should be some more current references than McMurray 2000 for monitoring of atmospheric aerosol since this sentence is about current key applications. One potential general one is Rose et al., 2021: https://acp.copernicus.org/articles/21/17185/2021/

**Answer: We will add this reverence**

L44 need space: "platforms(Petzold" change to "platforms (Petzold"

**Answer: ok**

L46 need comma: "2019) indoor" change to "2019), indoor"

**Answer: ok**

L53-54 "are subsequently acting" change to "subsequently act"

**Answer: ok**

L54-55 "are growing...magnitude" change to "grow by several orders of magnitude to a detectable size"

**Answer: ok**

L56 "are passing" change to "pass"

**Answer: ok**

L67 "to avoid" change to "of avoiding"

**Answer: ok**

L68 "Disadvantageously, water" change to "The disadvantages of water are that it"

**Answer: ok**

L70 "its likelihood" change to "the likelihood"

**Answer: ok**

L70 "inactivism" change to "inactivity"

**Answer: ok**

L70-71 "Activation of particles is hindered, e.g. by aerosol properties like water solubility and lipophilicity." change to "Further with water as the operating fluid, the activation of particles can be hindered, e.g. by aerosol properties like water solubility and lipophilicity."

**Answer: Great suggestion**

Line 80 "of German" change to "of the German"

**Answer: ok**

Line 84 "IAGOS aerosol instrument," change to "IAGOS aerosol instrumentation package"

**Answer: ok**

L84-90 I would move the first sentence of this paragraph to be the last sentence of this paragraph for better flow.

**Answer: ok**

L94 "Its final" change to "The reaction's final"

**Answer: ok**

L97 "it does not" change to "DMSO does not"

**Answer: ok**

L99-100 "An effect on measuring stations, that observe DMSO and DMSO products, is considered as minimal as the amount of DMSO released in the environment is far less, than the natural production." change to "For stations that measure DMSO and DMSO products, the effect of using DMSO as the working fluid in a CPC is considered minimal as the amount of DMSO released to the environment is far less than natural production."

**Answer: ok. Nevertheless, we add a phrase for the proximity dependence**

L104 - counters' change to counter

**Answer: ok**

L105 - "ratio, which" change to "ratio which"

**Answer: ok**

L107 - "like butanol" change to "to butanol"

**Answer: ok**

L108 - "restriction, that" change to "restrictions that"

**Answer: ok**

L113 - "to even -100 C" change to "to -100 C"

**Answer: ok**

L116 - delete "and which can still be detected" (Kelvin equation allows you to determine what particles will grow, but is not what determines detection).

**Answer: thanks**

L118 - number equations and refer to equation numbers in text. Make clear what M is molecular weight of - solute or solvent. Define p/ps.

**Answer: sure**

L120 - "The surface" change to "the surface"

**Answer: ok**

L124-125 "The Kelvin diameter shows from which saturation rate p/ps is in equilibrium with his surroundings and the droplet neither starts to grow nor shrink to due condensations processes or evaporating." change to "The Kelvin diameter is the droplet diameter at saturation rate  p/ps at which the droplet is at equilibrium and neither starts to grow nor shrink to due condensations processes or evaporation."

**Answer: ok**

L128 - capitalize Kelvin

**Answer: ok**

L128-129 "(Hinds, 1999)" change to "Hinds (1999)"

**Answer: ok**

L130 - explain difference between M and Ms in equation.

**Answer: ok (Different Material)**

L132 - "dissolved" change to "dissolve"

**Answer: ok**

L137 - "Temperature" change to "temperature"

**Answer: ok**

L137-138 "For this is to be solved for two different temperatures, the ratio of it will be the supersaturation for this calculated temperature difference." change to "Solving equation X for two temperatures and taking the ratio of the two results yields the supersaturation for this calculated temperature difference."

**Answer: thanks**

L144 - Caption figure 1 "Saturation ratio of different working fluids in dependence of the droplet diameter with a starting diameter of sodium chloride of 13 nm (Hinds, 1999)." change to "Saturation ratio of different working fluids as a function of the droplet diameter with a sodium chloride starting diameter of 13 nm (Hinds, 1999).'

**Answer: ok**

L147 - give equation number for Kelvin Kohler equation

**Answer: ok**

L150-152 - "As the particles pass the critical diameter as the critical saturation ratio is exceeded in their vicinity, they will grow in non-equilibrium if they face saturation ratios above 1 condition, which is defined as supersaturation." change to "If the critical saturation ratio is exceeded in the particle vicinity the particle will grow past the critical diameter.  If the saturation ratio is above 1 (i.e., supersaturation) then the particles will experience non-equilibrium growth."

**Answer: great suggestion**

L155 - don't capitalize butanol or water

**Answer: ok**

L156-157 - "Thus e.g. for particles size of 6 nm a saturation ratio of 1.037, and for 3 nm a saturation ratio of 1.078 has to be exceeded for DMSO." change to "For example, for DMSO, particles with diameters 6nm and 3nm require saturation ratios of 1.037 and 1.078, respectively.

**Answer: ok**

Figure 2 - legend - change Wasser to Water

**Answer: sure**

L160 - Caption Figure 2 - need space between "equations(Antoine"

**Answer: ok**

L163 - "while it is 0.026" change to "while the slope is 0.026"

**Answer: ok**

L166 - delete "in this work"

**Answer: ok**

L172 - capitalize Faraday

**Answer: ok**

L173-175 - "One Sky CPC 5.411, which from now on we designate as B-CPC to clarify that it was operated with butanol as its working fluid, and a second Sky CPC 5.411, which we will refer to as D-CPC to highlight that it was 175 operated with DMSO." change to "There are two Sky CPC 5.411 instruments designated B-CPC and D-CPC to indicate operating fluids of butanol and DMSO respectively."

**Answer: great suggestion**

L176 - "performed" change to "operated'

**Answer: ok**

L185 - "to smoothen the pressure regulation controlled" change to "to smooth the pressure oscillation generated"

**Answer: ok**

L187 - "controller" change to "controllers"

**Answer: ok**

L189 - "which is limited to" change to "which otherwise is limited to" note: I think what you are saying is that the mixing chamber RH is <30% unless a humidified air stream is added.

**Answer: With the added humidified air flow, the RH is limited to 30%**

Figure 3 - critical is spelled wrong in the schematic. Maybe label the CPCs: B-CPC, D-CPC and CPC 3772?

**Answer: oh… we redo this figure**

L199 - "to different particle sizes" change to "to a different particle size"

**Answer: ok**

L217-218 - "This resulted in a maximum particle mobility size of 138 bnm on the upper end." change to "This resulted in a maximum particle mobility size of 138 nm on the upper end for the salts and XXX nm for the soot."

**Answer: This maximum particle mobility size is for both particle types. We rephrased it a bit**

L221-222 - "leaving the DMA with the same mobility as the desired singly charged ones." change to "leaving the DMA with the same mobility as the singly charged particles."

**Answer: ok**

Figure 5 - threshold is spelled wrong in schematic

**Answer: ok**

L231 - don't capitalize multiple

**Answer: ok**

L231 - "scheme, that" change to "scheme that"

**Answer: ok**

L235 - "simply dividing through the C1/C0 value" dividing through what?

**Answer: The particle number concentration that is reported is divided by the C1/C0 value. The reported particle number concentration is the number of particles, that deliver a scattering signal when exceeding both detection threshold levels C1 and C0. By dividing the reported number concentration by C1/C0, the number concentration is adjusted to the C0 threshold, corresponding to smaller droplets.**

L235-236 - delete "This value was reported by the Sky CPC 5.411 as an internal quality parameter." repetitive with first sentence of paragraph

**Answer: ok**

L237 - "illustrated, that can surpass certain detector thresholds" change to "illustrated, along with   detector thresholds C0 and C1"

**Answer: ok**

L237 - again unclear what is being divided by C1/C0.

**Answer: see Answer to L235**

L243-244 - "The diameters of the DMSO droplets, which correspond to the signal heights of C0 and C1, are 2.5 – 2.9 um and 4.0 – 4.6 um." this is a unclear - where do the 2.9 and 4.0 come from?

**Answer: Those are the "uncertainties/ranges" of the droplet size for the detector thresholds. Those sizes depend on the refractive index of the droplet. One of these ranges is the one we calculated for PSL particles and the other for DMSO droplets**

L252 - uses Dp for diameter. earlier equations used lower case d for kelvin diameter. perhaps use D and Dp when talking about particle diameters?

**Answer: ok**

L259 "spare DMSO" what is spare DMSO?

**Answer: We had some amount of DMSO left over from other experiments that we did prior to these working fluid experiments. We have rephrased this.**

L259-261 "During initial DMSO experiments we started with a completely dry Sky CPC 5411. Prior to filling with DMSO, we performed a test run in the measurement setup and the dry CPC reported zero counts. The instrument wick was then wetted with DMSO and worked as intended. The CPC internal fluid controls operated as they do with butanol. We operated the D-CPC with the same parameters as the structurally identical B-CPC which was also in the experimental set-up (Fig. 3)."

**Answer: Great suggestion**

L266 "the C1/C0' change to "the C1/C0 value"

**Answer: ok**

L267-269 move "The C1/C0 reports the ratio of counts that have a higher and a lower detection threshold, used as internal CPC check for sufficient particle growth (described in Fig 6.)." to be the second sentence of the paragraph starting on line 234.

**Answer: ok**

L274-275 - "from the Sky-CPC for DMSO values are corrected by the division with the C1/C0 value to report all particles at the lower counting threshold." change to "from the D-CPC are corrected by dividing by the C1/C0 value in order to report all particles at the lower counting threshold."

**Answer: ok**

L275 - "The very first results were taken with" change to "The initial results were done with"

**Answer: ok**

L277 - "size, where" change to "size where"

**Answer: ok**

L277-278 "In Figure 7 it is visible, that the DMSO shows identical" change to "Figure 7 shows that the D-CPC has an identical D50 cut-off diameter as the B-CPC."

**Answer: ok**

L282 - "new working fluid" change to "DMSO"

**Answer: ok**

L287 - "Nevertheless, we dried" start new paragraph and change to "To start, we dried"

**Answer: ok**

L288 - "till no particle count reported and applied the new working-fluid as well." change to "until no particle counts were reported. We then added the DMSO."

**Answer: ok**

L289 - "equivalent to the DMSO Sky-CPC" change to "to be equivalent to the D-CPC"

**Answer: ok**

L289 - add the following sentence after "...well.": We will refer to the 3772-CEN using DMSO as D-CEN.

**Answer: good advice**

L293 - "As visible in Figure 8 counting for all CPC" change to "As shown in Figure 8, counting for all CPCs"

**Answer: ok**

Figure 8b - alternate legend: D-CEN 700, B-CPC 700, D-CEN 250, B-CPC 250, so it is consistent with figure 8a

**Answer: ok**

L299-300 - "The counting efficiencies do not differ as is visible in Figure 9 like in the previous experiments." change to "As in the previous experiments, the counting efficiencies for the different CPCs do not differ (Figure 9)."

**Answer: ok**

L301-302 - "We repeated all experiments using a mixture of 10% volumetric water to 90% DMSO (DW-CPC) does not influence the measurements in any way see Figure 9." change to "We repeated all experiments using a mixture of 10% volumetric water to 90% DMSO in the Sky CPC 5.411 (points labeled DW-CPC in Figure 9a). This mixture does not influence the measurements."

**Answer: ok**

L303-304 - "250 hPa, indicating, that water is evaporated and influences the particle growth." change to "250 hPa, indicating that water is evaporating and influencing the particle growth."

**Answer: ok**

L305 - change "the environmental operational conditions range" to "the lower range of operating conditions"

**Answer: ok**

L306 - delete "The pure DMSO substance has a melting point of 18°" already say this on line 113

**Answer: ok**

Figure 9 - please change fig 9a legend to alternate DW-CPC and B-CPC.  please change figure 9b legend to alternate B-CPC and D-CEN

**Answer: ok**

L309 caption for fig 9 - please change to "Counting efficiency curves with respect to the electrometer reference instrument and corrected for multiple charges at different operating pressures using fresh combustion soot particles for (a) Sky CPC 5.411 operated with butanol (B-CPC) and mixture of DMSO and water (DW-CPC) and (b) Sky CPC 5.411 operated with butanol (B-CPC) and the CPC 3772-CEN operated with DMSO (D-CEN)."

**Answer: ok**

L313 - "Figure 9" change to "Figure 9b"

**Answer: ok**

L314 - "measurement experience a sudden increase. This need further investigation and could not been" change to "measurement exhibits a sudden increase. This needs further investigation because it could not be"

**Answer: ok**

L316 - "We tested the substance in our measurement set up at various conditions, as such we increased the inline humidity to 30%, as we normally measure below 5% Even then," change to "We tested DMSO in our measurement set up at various conditions.  For example, we increased the inline humidity (RH) to 30%, although we normally measure at RH below 5%. Even then,"

**Answer: ok**

L318-319 "Regarding the cut-off diameter for D50 and D90 the Instruments do not differ significantly." change to "The instrument cut-off diameters for D50 and D90 do not differ significantly from each other regardless of particle type, operating fluid or pressure level."

**Answer: ok**

L319 - "for the cut-off" change to "for the D90 cut-off"

**Answer: ok**

L320 - "lowest pressure level at 200 hPa and 250 hPa." change to "lowest pressure level studied (either 200 hPa or 250 hPa, depending on experiment)."

**Answer: ok**

L321 - Table 1 caption - please change to "Table 1. Coefficients of the exponential fit of the counting efficiency curves and the D90 cut-off diameter (at either 200 or 250 hPa) for different particle types for the Sky CPC 5.411 operated with Butanol (B-CPC) and DMSO (D-CPC) and CPC 3772-CEN operated with DMSO (D-CEN)."

**Answer: ok**

Table 1 - why not include DW-CPC in Table 1?

**Answer: actually... the value for D-CPC soot are the values for DW-CPC, so we will correct this accordingly**

L326 - "5 days in a row" change to "5 consecutive days"

**Answer: ok**

L326-327 "with the Sky-CPC measuring the same number of particles."  change to "both the D-CPC and the B-CPC measured the same particle concentrations (Figure 10a)."

**Answer: ok**

L328-333 - change to "Since the overall ambient humidity ranged between 60 and 80%, it is certain that water condensed at the condenser and dropped into the wick of the 3772-CEN, weakening the activation efficiency (i.e., C1/C0). This is seen in Figure 10b, where the D-CEN trace is lower than the B-CPC trace starting around minute 45.  Because DMSO appears to affect the drain and priming valve gaskets the instrument needed to be manually drained (minute 60 to minute 80 in Figure 10b). After that, the 3772-CEN returns to the same concentration levels as the B-CPC." and start a new paragraph

**Answer: The C1/C0 adjustment does not apply to the 3772 CEN. The 3772 does not report those values, only the Sky-CPC does. "The CPC 3772-CEN has a built-in pulse height analyzer that triggers an error when the pulse amplitude decreases significantly below a minimum pulse height of 1 V. This automatically sets an error flag to indicate the presence of a measurement problem."**

Line 334-341 - I've suggested a re-write of these lines, but there are also some problems with the paragraph which I list after my suggested re-write.  "For the measurements shown in Figure 10 we recorded the working fluid consumption rate: the D-CPC was filled with 5 ml of DMSO, whereas the second B-CPC was filled with 20 ml butanol. We ran the measurements overnight. The B-CPC ran dry in the late morning (see green curve in Fig. 10b), the D-CPC was still running and counting. For the first set of ambient measurements, the C1/C0 value of the D-CPC stayed at 1 and decreased slowly until it oscillated around 0.8 (not shown). In a second set of ambient measurements, we added a water/DMSO mixture to a Sky-CPC (DW-CPC).  In this case the C1/C0 value decreased towards 0.6 slowly before increasing to XXX.

In Figure 8 the D-CPC number concentration was corrected with using the C1/C0 value, which was ranged around 0.8 for most (all?) of the measurement period shown."

(1) the legend in fig 10b says the green points are for the D-CEN so it doesn't make sense to look at the green line for the B-CPC running dry. The black line is for B-CPC but I don't see an obvious time where it was running dry and the D-CPC was not.

**Answer: You were right. This is clearly an error. I changed the picture in between and did not correct this statement.**

(2) the time axis in both figures is in min so referencing the time to 'late morning' is not meaningful. Perhaps say after XXX hours (i.e., at XXX min) on figure 10x.

**Answer: The event that a CPC has been "running dry" is not shown in this figure. We will rephrase the statement**

(3) the paragraph prior to my re-write referred several times to the C1/C0 value and referenced figure 10 but figure 10 does not show C1/C0 values.

**Answer: We will add RH and C1/C0 values to Figure 10a**

(4) say what the C1/C0 increased to in the DW-CPC case.

**Answer: 0.8 Again**

(5) the last sentence references figure 8. Should it? or did you mean figure 10?

**Answer: It should reference to fig 10**

Figure 10 - figure 10b legend should say D-CEN not CEN. (unless the legend is completely incorrect and the green is a different instrument (B-CPC?). If that's the case then discussion of earlier (lines 331-332) needs to be clarified.

**Answer: D-CEN it is**

L343 Caption Fig 10 - "Figure 10. Ambient air as measured by the Sky CPC 5.411 operated with (a) Butanol (B-CPC) and DMSO (D-CPC) as well for the (b) CPC 3772-CEN operated with DMSO (D-CEN), that needed manual extraction of condensed water, which caused the number concentration to drop." change to "Figure 10. Ambient air as measured by (a) the Sky CPC 5.411 operated with butanol (B-CPC) and DMSO (D-CPC) and (b) the B-CPC, D-CPC and CPC 3772-CEN operated with DMSO (D-CEN). D-CEN needed manual extraction of condensed water, which caused the number concentration to drop (see discussion in text)."

**Answer: Acknowledged**

L346-347 - "As is visible in Figure 11 measurements using DMSO working fluid is not distinguishable from the ones performed with Butanol." change to "Figure 11 demonstrates that the measurements made using DMSO as the working fluid are not distinguishable from the measurements performed with butanol."

**Answer: ok**

Figure 11 - what is the "G-CPC" mentioned in the legend - did you mean "B-CPC"?

**Answer: Yes, originally, we labeled it G-CPC (Grimm-CPC), but later renamed it to B-CPC. I simply overlooked it there. My fault.**

L353 Caption Figure 11 - "Figure 11. Comparison of the concentration linearity of two units Sky CPC 5.411 to the electrometer reference at different pressure levels for ammonium sulphate (a) and ambient air (b)." change to " Figure 11. (a) Comparison of the concentration linearity of two Sky CPC 5.411 units to the electrometer reference at different pressure levels for ammonium sulphate and (b) comparison of correlation between B-CPC and D-CPC on ambient air." (there is no electrometer data on figure 11b.)

**Answer: Thanks**

L356-357 - "As already mentioned, the CPC 3772-CEN needed manual draining and the Sky CPC 5.411 needed manual priming."  it was unclear on L328 that the priming issue was related to the Sky CPC. Please clarify on L328.

**Answer: The priming / filling – issue is correct for both CEN and Sky-CPC. On both instruments we experience the issue with the blocked valve. We will rephrase this.**

L358, L360, L363 - "O-rings" change to "o-rings"

**Answer: ok**

L362-364 - "We repeated the process with a third Sky CPC 5.411 and it performed the same test as in the first experiments, including the soaked O-ring." change to "We repeated the process using DMSO as the operating fluid in a third Sky CPC 5.411 - this unit performed the same as the D-CPC in the first experiments, including agreeing well with the B-CPC and the swelled o-ring."

**Answer: ok**

L368 Caption Figure 12 - "Figure 12. Picture of the gasket of a butanol CPC and a DMSO CPC (left), used in butanol (a) and DMSO (b) as well as of the stem of the liquid supply valve (right)." change to "Figure 12. Picture of the CPC o-ring (left, top and bottom), and the stem of the liquid supply valve (right). (a) indicates operating fluid was butanol and (b) indicates operating fluid was DMSO."

**Answer: ok**

L372-375 - "We tested for several weeks atmospheric aerosol and did not observe any significant shift between the reference butanol CPC. However, higher ozone or nitrogen oxides concentrations could trigger secondary organic aerosol activation within the instrument." change to "We measured atmospheric aerosol for several weeks using instruments with DMSO as the operating fluid. We did not observe any significant shift in measured particle number concentrations between the reference B-CPC and the D-CPC. However, higher ozone or nitrogen oxides concentrations could trigger secondary organic aerosol activation within the instrument using DMSO."

**Answer: thanks**

L377-378 - "It is not conclusive that this effect should also be attributed to the use of DSMO, as the instrument's maintenance state or operational errors could also have been factors." change to "It is not clear that this issue can be attributed to the use of DSMO, as the instrument's maintenance state or operational errors could also have been factors."

**Answer: ok**

L380 "Therefore, it should be avoided to be disposed into the sewer (Tucker, 2014)." change to "Therefore, DMSO should not be disposed of into the sewer (Tucker, 2014)."

**Answer: ok**

L385 "We are not certain if the signal is generated by droplets or solid crystals, because the freezing point of DMSO is below the condenser temperature." This sentence is kind of a non sequitur. Perhaps it would be better earlier in the paper - e.g., on L167. Or you could start a new paragraph on line 390 since it seems to be related to the condensation process sentence.

**Answer: ok**

L395 - "use also from an instrument" change to "use, from a user or an instrument"

**Answer: ok**

L409 - "counters (CPC), which is an" change to "counters (CPC). DMSO is an"

**Answer: ok**

L430 - "to the fact, that" change to "to the fact that"

**Answer: ok**